# Dietary Lactoferrin Supplementation Improves Growth Performance and Intestinal Health of Juvenile Orange-Spotted Groupers (*Epinephelus coioides*)

**DOI:** 10.3390/metabo12100915

**Published:** 2022-09-28

**Authors:** Tao Song, Yingmei Qin, Liner Ke, Xuexi Wang, Kun Wang, Yunzhang Sun, Jidan Ye

**Affiliations:** 1Xiamen Key Laboratory for Feed Quality Testing and Safety Evaluation, Fisheries College of Jimei University, Xiamen 361021, China; 2Key Laboratory of Marine Biotechnology of Fujian Province, College of Marine Sciences, Fujian Agriculture and Forestry University, Fuzhou 350002, China

**Keywords:** lactoferrin, *Epinephelus coioides*, growth performance, intestinal damage

## Abstract

A 56-day feeding trial was conducted to investigate the effects of dietary lactoferrin (LF) supplementation on the growth performance and intestinal health of juvenile orange-spotted groupers fed high-soybean-meal (SBM) diets. The control diet (FM) and high-soybean-meal diet (SBM60) were prepared to contain 480 g/kg protein and 110 g/kg fat. Three inclusion levels of 2, 6, and 10 g/kg LF were added into the SBM60 to prepare three diets (recorded as LF2, LF6, and LF10, respectively). The results showed that the supplementation of LF in SBM60 increased the growth rate in a dose-dependent manner. However, the feed utilization, hepatosomatic index, whole-body proximate composition, and the abundance and diversity of intestinal microbiota did not vary across the dietary treatments (*p* > 0.05). After the dietary intervention with LF, the contents of the intestinal malondialdehyde, endotoxin, and d-lactic acid, as well as the plasma low-density lipoprotein cholesterol, high-density lipoprotein cholesterol, and total cholesterol were lower, and the intestinal activities of the glutathione peroxidase, lipase, trypsin, and protease were higher in the LF2-LF10 groups than that in the SBM60 group (*p* < 0.05). The supplementation of LF in SBM60 increased the muscle layer thickness of the middle and distal intestine and the mucosal fold length of the middle intestine vs. the SBM60 diet (*p* < 0.05). Furthermore, the supplementation of LF in SBM60 resulted in an up-regulation of the mRNA levels for the *IL-10* and *TGF-β1* genes and a down-regulation of the mRNA levels of the *IL-1β*, *IL-12*, *IL-8*, and *TNF-α* genes vs. the SBM60 diet (*p* < 0.05). The above results showed that a dietary LF intervention improves the growth and alleviates soybean meal-induced enteritis in juvenile orange-spotted groupers. The dietary appropriate level of LF was at 5.8 g/kg, through the regression analysis of the percent weight gain against the dietary LF inclusion levels.

## 1. Introduction

Fish meal (FM) has always been the main important protein source in the feeds of farmed marine fish due to its high-quality protein, balanced amino acids profile, and less anti-nutritional factors (ANFs) in comparison with terrestrial animals and plant protein sources [1]. However, over the past three decades, the stagnant global marine fishery catches have also led to stagnant FM production due to the shift in climate change [2]. At the same time, the rapid expansion of the aquaculture industry has stimulated a huge demand for FM, resulting in a sustained shortage of FM supply [3]. Therefore, research into FM replacement with other protein sources has always been one of the most priority issues in aquafeeds. Soybean meal (SBM), as the major plant protein source, is widely used in aquafeeds due to its relatively balanced amino acids profile, huge output and availability, and reasonable price [1,4,5]. However, when SBM is used in aquafeeds in a large proportion, its side effects are also considerable. It is well known that SBM can cause the reduction in the ability of fish to digest and absorb nutrients, due to many ANFs that induce enteritis [6,7], the so-called soybean meal-induced enteritis (SBMIE) [8]. Therefore, how to prevent and control the widespread SBMIE is the key to maintain a normal daily fish culture and reduce disease risk.

In the last two decades, a great deal of research effort has been devoted to alleviating or/and preventing SBMIE. An effective strategy to counteract fish intestinal inflammation is the dietary use of functional substances that exert metabolic regulation, antioxidant properties, and immune promotion [4,9,10,11,12]. Lactoferrin (LF), a glycoprotein with a molecular weight of 78–80 kDa, is rich in milk [13,14] and has many biological functions [14,15,16]. A previous study showed that dietary LF supplementation could enhance the growth performance of early-weaned piglets by promoting the proliferation of intestinal beneficial bacteria, such as lactic acid bacteria; inhibiting the proliferation of *E*. *coli*; and improving the intestinal mucosal morphological structure and intestinal function [17]. Dietary LF supplementation could improve the growth performance of early-weaned piglets and zebrafish by reducing pathogenic bacteria and diarrhea, enriching beneficial bacteria, and affecting intestinal morphology [15,16]. LF was found to inhibit the activation of the TOR signaling pathway induced by lipopolysaccharide (LPS), thereby inhibiting the production of pro-inflammatory factors such as *IL-1β* and *IL-8* or/and indirectly promoting the production of anti-inflammatory factors such as *IL-10* in terrestrial animals [18]. However, little is known about the dietary LF intervention effect on the improvement of SBMIE in farmed fish.

An orange-spotted grouper (*Epinephelus coioides*) is a marine economic carnivorous fish, which is widely cultured in Southeast Asian countries, including China [19,20]. According to the China Fishery Statistics Yearbook (2021), this fish has become the third largest mariculture fish in China, with an annual output of 192,045 tons in 2020 [21]. Although there are many reports about its nutrition, feed research, and development [20,22,23], there is still a lack of nutritional regulation research on the prevention of SBMIE in the fish species. The recent studies also showed that a high-SBM diet caused juvenile orange-spotted grouper enteritis [1,5]. Therefore, in this study, the growth performance, intestinal health, and prevention or/and control of SBMIE were investigated to evaluate the effects of dietary LF supplementation in juvenile *E. coioides* fed high-SBM diets. For this purpose, five experiment diets were prepared, including the FM diet, SBM60 diet without LF, and three SBM60 diets with LF at inclusion levels of 2, 6, and 10 g/kg. This study provided a new technical reference for the prevention of SBMIE in the fish species.

## 2. Materials and Methods

### 2.1. Experimental Diets

A control diet (FM) was formulated using FM, casein, and gelatin as the protein sources and fish oil, soybean oil, and soybean lecithin as the lipid sources to contain 480 g/kg crude protein and 110 g/kg crude lipid (Table 1). On the basis of the FM diet, SBM was used to replace 600 g/kg FM protein to prepare a high-SBM diet (SBM60). LF was then added to SBM60 diets at 2, 6, and 10 g/kg to prepare another three experimental diets (LF2, LF6, and LF10, respectively), according to a previous study [17]. The coarse dry feed ingredients were pulverized using a grinder (ZFJ-300, Jiangyin Ruizong Machinery Manufacturing Co., Ltd., Jiangyin, Jiangsu, China) and sifted through a 60-mesh sieve (250 μm particle size), weighed and homogenized. The liquid ingredients (water, soybean oil, fish oil, and soy lecithin) were then added to the dry feed ingredients and a mash was prepared. This dough was extruded into strands and pelletized through 2.5 and 4 mm die using cold press extrusion (F-76, Guangzhou Huagong Optical Mechanical and Electrical Technology Co., Ltd., Guangzhou, Guangdong, China) and a feed pellet shaping machine (GY-500, Changzhou Beicheng Drying Equipment Engineering Co., Ltd., Changzhou, Jiangsu, China). The pellets were dried in a ventilated oven at 55 °C for 24 h to reduce the moisture of the feed to less than 100 g/kg and then stored at room temperature for 24 h, before being sealed in plastic bags and stored in a refrigerator at −20 °C.

### 2.2. Feeding Trial

This experiment was conducted at Fujian Dabeinong Fisheries Technology Company (Zhaoan County, Zhangzhou City, Fujian, China). Prior to the trial, orange-spotted grouper juveniles were kept in a concrete pond and fed with diet FM for a 2-week acclimatization. At the beginning of the experiment, 450 orange-spotted grouper juveniles (initial average weight of 33.82 ± 0.03 g) were randomly distributed into 15 blue polypropylene tanks (500 L/tank), at a density of 30 fish per tank in a water temperature-controlled recirculating culture system. Groups of triplicate tanks were hand fed one of the diets to apparent satiation twice daily (8:00 and 17:00) under a natural photoperiod across a feeding period 56 days. Excess feed was collected 30 min after each meal and dried for 24 h at 65 °C and weighed for the calculation of feed intake. Because of the water loss of the aquaculture system caused by daily sewage discharge, fresh sea water was refilled until the original water level of tanks was reached. During the experimental period, water salinity ranged from 32 to 36, temperature was about 28.5 °C, and dissolved oxygen level was >5.7 mg/L.

### 2.3. Sample Collection and Chemical Analysis

At the end of the growth trial, fish in each tank were caught and anaesthetized with a dose of 100 μḷ/L solution of eugenol (Nanjing wensenbao International Trade Co., Ltd., Nanjing, Jiangsu, China). Fish weight and number were then recorded for each tank to measure weight gain (WG), feed efficiency (FE), specific growth rate (SGR), and survival. Three fish from each tank were randomly sampled and pooled in plastic bags and stored at −20 °C for whole-body proximate composition determination. A total of 9 fish per tank (27 fish each group) were weighed individually after anesthesia with eugenol (100 μḷ/L) to calculate the hepatosomatic index (HSI) and condition factor (CF). Blood was drawn from the caudal vein, using 1 mL heparinized syringe, and centrifuged at 1027× *g*, 4 °C, 10 min. Plasma was then collected, pooled by tank, and stored in 1.5 mL Eppendorf tubes at −80 °C for the subsequent biochemical analysis. The intestines of nine fish per tank were aseptically removed and pooled into one tube by tank, stored at −80 °C for the analysis of biochemical components, microbiota analysis, and gene expression.

Prior to component analysis, whole-fish samples were prepared according to the method described by Ye et al. [24]. The proximate composition of diet and whole-body fish samples were determined according to standard methods [25]. Dry matter was determined by drying the samples in an oven at 105 °C to a constant weight. Crude protein was determined by the Kjeldahl method (*N* × 6.25), using Kjeltec TM 8400 Auto Sample Systems (Foss Teacher AB). The crude lipid content was determined by the Soxtec extraction method by using Soxtec Avanti 2050 (Foss Teacher AB). Ash was measured in the residues of samples burned in a muffle furnace at 550 °C for 8 h.

The activities of intestinal protease, trypsin, and amylase were determined according to the method described by Hu et al. [26]. Casein was used as the reaction substrate, and the reaction product was determined by Folin’s reagent. At 37 °C, intestinal protease degraded casein for 1 min to produce 1 μg tyrosine as an enzyme activity unit, expressed as U/mg protein. *Na*-benzoyl-arginine-*p*-nitroanilide (BAPNA) was used as the reaction substrate. Degradation of BAPNA by the trypsin for 1 min at 37 °C resulted in 1 μmol *p*-nitroanilide as an enzyme activity unit, expressed as U/g protein. Starch was used as the reaction substrate, and the degradation of 10 mg of starch by amylase at 37 °C at 30 min was taken as an enzyme activity unit, expressed as U/mg protein. The total cholesterol (TC), triglycerides (TG), high-density lipoprotein cholesterol (HDL-C), and low-density lipoprotein cholesterol (LDL-C) contents in plasma samples; the contents of diamine oxidase (DAO), d-lactic acid (D-Lac), endotoxin (ET), endothelin-1 (ET-1), glutathione peroxidase (GSH-Px), superoxide dismutase (SOD), total antioxidant capacity (T-AOC), and malondialdehyde (MDA) in the intestinal samples; and the activity of lipase in the intestinal samples were determined using commercial kits (Nanjing jiancheng Bioengineering Institute, Nanjing, Jiangsu, China), according to the manufacturer’s instructions.

### 2.4. Intestinal Histology Observation

One fish was caught from each tank and dissected to obtain the whole gut, then divided into proximal, middle, and distal intestine (i.e., PI, MI, and DI, respectively), according to the method described by Anguiano et al. [27]. All the segments were washed with normal saline, fixed in Bouin’s solution for 24 h, rinsed with 70% (*v*/*v*) ethanol solution, and finally immersed in 70% (*v*/*v*) ethanol until histological processing was performed [19]. The fixed gut segments were embedded in paraffin and 5 µm sections were cut by using a rotary microtome (KD-2258S, China). The serial histological sections were then mounted on glass slides and stained with hematoxylin and eosin for morphometric analysis. Pictures were examined under a light microscope (Leica DM5500B, Germany), and digital images were taken and processed with a digital camera (Leica DFC450) equipped with the image program LAS AF (Version 4.3.0 Leica). Five slides were prepared for each gut segment sample and 30 measurements were made to determine the number of mucosal folds, muscle layer thickness, and length of the complete mucosal fold.

### 2.5. Intestinal Microbiota Analysis

The total DNA in the distal intestine (DI) of juvenile orange-spotted groupers were extracted using a DNA extraction kit (Omega Bio-teK, Norcross, GA, USA) according to the manufacturer’s instructions. The integrity and quality, purity, and quantity of DNA samples were assessed by electrophoresis on a 1% (*w*/*v*) agarose gel and spectrophotometer method (NanoDrop 2000, Wilmington, DE, USA 260 nm/280 nm optical density ratio), respectively. The V3-V4 region of the 16S rDNA gene of DI bacterial was amplified by polymerase chain reaction (PCR) using the forward primer 338F (5′-ACTCCTACGGGAGGCAGCAG-3′) and the reverse primer 806R (5′-GGACTACNNGGGTATCTAAT-3′). The PCR reaction system included pre-denaturation at 95 °C for 5 min; denaturation at 95 °C for 45 s, annealing at 55 °C for 50 s, and extension at 72 °C for 45 s, 32 cycles; extension at 72 ℃ for 10 min. Subsequently, high-throughput sequencing was performed using Illumina Miseq PE300 at Beijing Allwegene technology Co., Ltd. (Beijing, China). A library of small fragments was constructed using paired-end for sequencing, and the data were passed through QIIME (v1.8.0) for removal low-quality sequences and chimeras. Based on 97% sequence similarity, similar sequences were assigned to the same operational taxonomic units (OTU). Species classification information corresponding to each OTU was obtained by comparing with the sliva database, and alpha diversity analysis (Shannon, ACE, and Chao1) was performed using Mothur software (version 1.31.2). Based on the weighted unifrace distance, the pheatmap of R (v3.1.1) software package was used for clustering analysis. After the UniFrac algorithm, the information of system evolution was used to compare the difference in species communities among samples and Beta diversity analysis was performed.

### 2.6. RNA Extraction and Gene Expression

The total RNA was extracted from the intestinal samples using TRIzol^®^ reagent (Takara Co., Ltd., Tokyo, Japan) according to the manufacturer’s instructions. Isolated RNA was quantified using the NanoDrop ND-2000 Spectrophotometer, and its integrity was confirmed by agarose gel electrophoresis. The cDNA was generated from 1 μg DNase-treated RNA and synthesized by a PrimeScript^TM^ RT Reagent Kit with gDNA Eraser (Perfect Real Time) (Takara Co., Ltd., Tokyo, Japan). Real-time PCR was employed to determine mRNA levels based on the TB Green^TM^ Premix Ex Taq^TM^ Ⅱ (Tli RNaseH Plus) (Takara Co., Ltd., Tokyo, Japan) using a QuantStudio^TM^ Real-Time PCR System (ABI) quantitative thermal cycler. The fluorescent quantitative PCR solution consisted of 10 μL TB Green Premix Ex Taq^TM^ Ⅱ (Tli RNaseH Plus) (2×), 0.8 μL PCR forward primer (10 μM), 0.8 μL PCR reverse primer (10 μM), 2.0 μL RT reaction (cDNA solution), and 6 μL dH_2_O. The thermal program included 30 s at 95 °C, 40 cycles at 95 °C for 5 s, and 60 °C for 30 s. The sequences of primers are showed in Table 2. All amplicons were initially separated by agarose gel electrophoresis to ensure that they were of the correct size. β-actin served as the internal reference gene to normalize cDNA loading. The gene expression levels of the target genes were analyzed by the 2^−ΔΔCt^ method [28] after verifying that the primers were amplified with an efficiency of approximately 100% [29], and the data for all treatment groups were compared with the data for the control group.

### 2.7. Statistical Analysis

All data were presented as mean and standard error of the mean (SEM). The data were analyzed using a one-way analysis of variance (ANOVA) to test for differences between treatments and then the Student–Neuman–Keuls multiple comparison test was performed after confirming the normality and homogeneity of variance using the Kolmogorov–Smirnov test and Levene’s test in SPSS Statistics 25.0 (SPSS, Michigan Avenue, Chicago, IL, USA). The data expressed as percentages or ratios were subjected to data conversion prior to statistical analysis. *p*-value < 0.05 was deemed as significant difference.

## 3. Results

### 3.1. Growth Performance and Whole-Body Proximate Composition 

The results of the growth performance and whole-body proximate composition are presented in Table 3. The diets SBM60 had a lower WG and SGR compared with the diets FM (*p* < 0.05), but the diets LF2-LF10 showed an improved WG and SGR vs. the diets SBM60 (*p* < 0.05) and returned to the level of the diets FM (*p* > 0.05). The WG and FE were in a dose-dependent relationship with the dietary LF inclusion levels (Figure 1). The maximum WG and FE were observed for the diet LF6 and diet LF10, respectively. However, there were no differences in the FE, HSI, CF, survival, and whole-body proximate composition among the dietary treatments (*p* > 0.05).

### 3.2. Intestinal Antioxidant Capacity

As shown in Table 4, the FM group had higher intestinal GSH-Px activity and lower MDA content compared with the SBM60 group (*p* < 0.05), but the values of the two parameters were not different between the LF2-LF10 groups and the FM group (*p* > 0.05). The MDA content showed a negative quadratic response to the increasing dietary LF inclusion levels; a minimum value was observed for the diet LF6. However, the dietary treatments did not affect the intestinal SOD and T-AOC activities (*p* > 0.05).

### 3.3. Plasma Components

Table 5 shows that plasma HDL-C content had an irregular change with increasing dietary LF inclusion levels, and the value in the LF6 group was similar (*p* > 0.05) to that of the SBM60 and FM groups but was higher than that of the LF2 and LF10 groups (*p* < 0.05). The FM group had higher plasma LDL-C content (*p* < 0.05) and comparable plasma TC content compared with the SBM60 group (*p* > 0.05), but the plasma TC content was not different between the LF2-LF10 groups and SBM60 group (*p* > 0.05), while the SBM60 and LF2 groups had higher plasma LDL-C contents (*p* < 0.05) vs. the LF6 and LF10 groups. Both the plasma LDL-C and TC contents showed a linear decreasing trend with the dietary increase in the LF inclusion levels and reached the minimum values at LF6 and LF10, respectively. However, the plasma TG content did not differ across all the dietary treatments (*p* > 0.05).

### 3.4. Intestinal Digestive Enzyme Activity

As shown in Table 6, the intestinal lipase and protease activities were comparable, and the intestinal trypsin activity was lower in the SBM60 group than that in the FM group (*p* < 0.05). The intestinal trypsin and lipase activities were higher in the LF2-LF10 groups than that in the SBM60 group (*p* < 0.05) and returned to the level of the FM group and even higher than that of the FM group. The intestinal activities of the lipase, trypsin, and protease showed a linear increasing trend with increasing dietary LF inclusion levels. However, the intestinal amylase activity was not affected by the dietary treatments (*p* > 0.05).

### 3.5. Intestinal Permeability

Table 7 shows that the intestinal D-Lac and ET contents were higher in the SBM60 group than that in the FM group (*p* < 0.05). However, the intestinal D-Lac and ET contents were reduced when given the SBM60 diet with the LF supplementation (*p* < 0.05), and the values in the fish receiving the SBM60 diets with the LF supplementation returned to the level and even lower than that of the FM group. The intestinal D-Lac and ET contents did not differ with the dietary LF levels from 2 to 10 g/kg (*p* > 0.05). Both the intestinal ET-1 and DAO contents were not affected by the dietary treatments (*p* > 0.05).

### 3.6. Intestinal Histomorphology

Table 8 shows the effects of the dietary treatments on the mucosal fold number (nMF), muscle layer thickness (tML), and mucosal fold length (lMF) in the three intestinal segments (PI, MI, and DI). The nMF of the PI, MI, and DI; the lMF of the PI and DI; and the tMF of the PI remained unaffected by the dietary treatments (*p* > 0.05), but the tML of the DI showed a positive quadratic response to increasing dietary LF inclusion levels, with a maximum value observed for diet LF6. Diets LF2 to LF10 displayed a higher (*p* < 0.05) tML of the DI vs. the diet SBM60, and the value returned to the level of the diet FM (*p* > 0.05). The lMF and tML of the MI had an irregular change in response to the dietary LF inclusion levels, but the maximum values observed all for the diet LF2.

### 3.7. Abundance and Difference in Intestinal Microbiota

The Firmicutes, Bacteroidetes, and Proteobacteria in the DI were the dominant phyla of all the dietary treatments (Figure 2A). Compared with the diets FM, the abundance of Firmicutes was increased and the abundances of the Bacteroidetes and Proteobacteria were decreased by the diets SBM60. The abundances of the Firmicutes and Bacteroidetes generally increased, but the abundance of the Proteobacteria decreased with increasing dietary LF-levels supplementation in the SBM60 diets (Figure 2A and Appendix A). However, no significant differences in the abundances of the dominant phyla were observed between the dietary treatments at the phylum level (*p* > 0.05).

At the genus level, the DI bacteria of all the dietary treatments mainly contained the genera *Photobacterium*, *Selenomonas-1*, *Prevotella-1*, *Vibrio*, and *Rikenellaceae-RC9-gut-group* (Figure 2B and Appendix A). The decreased abundances of the genera *Photobacterium*, *Selenomonas-1*, and *Prevotella-1* and the increased abundances of the genera *Vibrio* and *Rikenellaceae-RC9-gut-group* were observed in the diets SBM60 vs. the diets FM. The abundances of the genera *Selenomonas-1*, *Prevotella-1*, and *Rikenellaceae-RC9-gut-group* showed an open downward parabola response, but the abundances of the genera *Photobacterium* and *Vibrio* showed an open upward parabola in response to the increasing dietary LF inclusion levels. However, there was no significant difference in the abundances of the dominant genera between the dietary treatments at the genus level (*p* > 0.05). 

### 3.8. Expression of Intestinal Inflammatory Factor Genes

As shown in Figure 3, the FM group had lower mRNA levels for the *IL-1β*, *IL-12*, *IL-8*, and *TNF-α* in comparison with the SBM60 group (*p* < 0.05), and no differences in the value for the *TGF-β1* and *IL-10* were observed between them (*p* > 0.05). Although the intestinal mRNA levels for the *IL-1β*, *IL-12*, *IL-8*, *TGF-β1*, and *TNF-α* did not differ across the diets from LF2 to LF10 (*p* > 0.05), the values for the *IL-1β*, *IL-12*, *IL-8*, and *TNF-α* were lower, but the value for the *TGF-β1* was higher in the diets LF2–LF10 than in the diet SBM60 (*p* < 0.05), and returned to the level of the FM group (*p* > 0.05) and even lower than that of the FM group. The mRNA level for the *IL-10* was not different in the LF2 and LF10 groups (*p* > 0.05) but higher than the FM, SBM60, and LF6 groups (*p* < 0.05).

## 4. Discussion

Our previous study and other studies have shown that high-SBM diets resulted in poor fish growth performance [1,5,30,31], which supported our current results that the growth rate was reduced when juvenile orange-spotted groupers were fed high-SBM diets vs. FM diets, as evidenced by a significant decrease in the WG and SGR. Consistent with our previous results [1], no variations between the high-SBM diets and FM diets in the HIS [31,32,33], CF [30,33,34], and FE [5] were observed. In contrast, feeding a high-SBM diet led to a reduction in the HSI, CF, and FE. The restricted growth of fish caused by high-SBM diets might be due to the imbalanced amino acids profile in high-SBM diets [5,31] and SBMIE, a resultant low feed utilization [1,35]. It is clear that dietary LF administration shows a strong effect on improving growth for terrestrial animals. For example, LF administration could enhance the growth performance of early-weaned piglets and neonatal calves [36,37]. Recently, Olyayee reported a growth promotion and carcass yield of broiler chicken when fed a diet with 0.8 g/kg LF [38]. In fish, LF was used as a feed additive to promote the growth of farmed fish [39,40,41]. However, in other studies, this promoting effect was not observed in Atlantic salmon [42], Nile tilapia [43], gilthead sea bream [44], African cichlid [45], grouper [46], and Siberian sturgeon [47]. This discrepancy may be ascribed to the differences in fish species and physiological status, growth stage, LF dosage, water temperature, etc.

There were no differences in the whole-body proximate composition between fish fed high-SBM diets and fish fed FM diets in the present study and previous studies with largemouth bass [31], rockfish [48], and pompano [49]. In contrast, previous studies showed that feeding high SBM resulted in an increase in moisture content [33,50] and a decrease in crude protein content [33,50] and crude lipid content [1,33,50]. After dietary LF intervention, no notable changes were observed in the whole-body proximate composition in the present study and a previous study on silvery-black porgy [51], which was inconsistent with El-Sayed and Al-Kenawy [40], who reported that dietary LF increased the whole-body protein content and decreased the whole-body lipid content of Nile tilapia.

The activities of digestive enzymes reflect the ability to digest feed [52]. In the present study, the intestinal activities of trypsin and protease were decreased in fish fed with high-SBM diets vs. fish fed FM diets, which supported the results in previous studies with drum fish [53], Japanese seabass [54], hybrid tilapia [55], as well as our previous study [1]. The reduced intestinal trypsin and the protease activities were the result of the presence of trypsin inhibitors or other ANFs in the SBM, resulting in a poor growth performance and feed utilization in fish [55,56,57,58]. After the dietary intervention of LF, the intestinal activities of the lipase, protease, and trypsin showed a linear response to the dietary LF inclusion levels and reached or even exceeded the original levels of the FM diet, indicating that LF could promote the digestion of feed. Dietary LF administration alone or in combination with other functional feed additives was also reported to have an enhancement effect on the intestinal trypsin and protease activities of silvery-black porgy [59,60]. Dietary LF administration could promote the proliferation of intestinal epithelial cells and protect the intestinal crypt and villous structure of piglets [61,62], reflecting the integrity of the intestinal mucosa and the stabilization of the intestinal brush border. Therefore, it is natural to improve the digestive capacity through the intervention of dietary LF on the enteritis response of fish.

The intestinal tract is not only the place where nutrients are digested and absorbed [63] but also a site of immune defense, an important barrier against exogenous pathogens [64]. D-Lac and ET are the products secreted by the inherent bacteria of the gastrointestinal tract, and increases in their concentrations in the blood reflect the dysregulation of the intestinal flora and impaired permeability [65,66,67,68]. In this study, the intestinal D-Lac and ET contents increased in the diets SBM60 vs. that in the diets FM. This finding was consistent with previous results [1,54,69], indicating the intestinal mucosal injury of fish induced by dietary high SBM. So far, there is no report on the intervention of LF in the SBMIE of fish. In the present study, we observed reduced intestinal ET and D-Lac contents in juvenile orange-spotted groupers administrated with dietary LF vs. those of fish fed SBM60 diets, reduced to the level of the control or even lower. This indicated that dietary LF supplementation in a high-SBM diet could reduce the intestinal mucosal permeability of the fish species.

An intact intestinal histomorphology is a prerequisite for maintaining a good digestion and absorption state and intestinal immune disease resistance [70]. The histomorphological indicators, such as the lMF, tML, and nMF, were used to evaluate the intestinal digestion and absorption capacity in previous studies [1,19]. In the present study, a significant reduction in the tML in the DI was observed in the SBM diets vs. the FM diets, as evidenced by a previous study [1]. Many studies showed that fish fed high-SBM diets had shortened villus height and thinned muscular thickness [5,54,71], which indicated a potential abnormal intestinal histomorphology caused by dietary high SBM. The intestinal histopathological changes were improved when fish were intervened by the dietary LF administration in our current study. We observed an increase in the lMF in the MI, and the tML in the MI and DI in a dose-dependent manner with increasing LF levels in high-SBM diets. Consistent with the results of our current study, dietary LF administration increased the height, width, and surface area of intestinal villi, the depth of the crypts, and the thickness of the muscular layer of broiler chickens [38]. The positive effect of dietary LF administration on the intestinal histomorphology of terrestrial animals has been widely recognized [72,73,74]. Therefore, the intervention effect of dietary LF administration on the SBMIE of juvenile orange-spotted groupers can also be achieved by improving the histomorphology.

The intestine is also the so-called “second brain” that links between the intestinal microbiota and diseases. Dysbacteriosis of the intestinal flora increases the susceptibility to intestinal pathogens, and in severe cases, it will further develop into intestinal infection and reduce immune function [75,76]. In the present study, fish fed high-SBM diets showed an increased abundance of the phylum Firmicutes and a decreased abundance of the phylum Bacteroidetes and Proteobacteria vs. the fish fed the FM diets. At the genus level, the fish fed high-SBM diets exhibited a decreased abundance of *Photobacterium*, *Selenomonas_-_1*, and *Prevotella_-_1* and an increased abundance of *Vibrio* and *Rikenellaceae_-_RC9_-_gut_-_group* vs. the fish fed FM diets. Nevertheless, a high-SBM substitution for FM did not alter the intestinal microbial abundance and diversity of groupers at either the phylum level or the genus level, which was inconsistent with previous studies in largemouth bass [77], gilthead sea bream [78], and large yellow croaker [79]. The difference in the relative abundance of intestinal microbiota at the phylum and genus levels may be directly or indirectly influenced by external environmental conditions, such as the living environment, fish species, growth stage, and feed composition. In the present study, the phylum Firmicutes, Bacteroidetes, and Proteobacteria were identified as the dominant phyla of the intestine in fish fed high-SBM diets with LF administration. The relative abundances of the Firmicutes and Bacteroidetes were generally promoted, while the abundance of the Proteobacteria was reduced by the intervention of the dietary LF administration. At the genus level, the intestinal microbiota diversity did not vary, but the relative abundance of the genus bacteria varied with the dietary LF inclusion levels. As mentioned above, thus far, there are few reports regarding the intervention of LF on fish enteritis.

The presence of an inflammatory response is a complex pathophysiological process, which is mediated by the activation of a variety of cytokines and complement factors secreted by macrophages and leukocytes [80]. LF can specifically bind B-lymphocytes and macrophages, thus inhibiting the production of pro-inflammatory factors, such as *IL-1β*, *IL-8*, and *TNF-α*, or/and indirectly promoting the production of anti-inflammatory factors such as *IL-10* through preventing the activation of the TOR signaling pathway induced by lipopolysaccharide [18,81,82]. LF was used to modulate the inflammatory response, with an emphasis on protection against intestinal infections and inflammatory bowel diseases of mammals [82,83,84]. Like mammals, fish have pro-inflammatory cytokines, including *IL-1β*, *TNF-α*, *IL-8*, and *IL-12*, and the anti-inflammatory ones are *IL-10* and *TGF-β1* in fish immune responses [1,85,86]. It is clear that the up-regulation of pro-inflammatory gene expression and the down-regulation of anti-inflammatory gene expression are caused by SBMIE [87,88]. In the present study, the fish fed with the high-SBM diets promoted the expression of such pro-inflammatory genes as *IL-1β*, *TNF-α*, *IL-8*, and *IL-12* vs. the fish fed with the FM diets. The findings agreed with what has been reported in previous studies with an SBM substitution for FM in different fish species [1,5,77]. In the present study, the mRNA levels of the *IL-1β*, *TNF-α*, *IL-8*, and *IL-12* genes were down-regulated, while the mRNA levels of the *IL-10* and *TGF-β1* genes up-regulated after the fish were fed high-SBM diets administrated with LF. This finding indicated that a dietary LF intervention could alleviate the SBMIE of fish through promoting the production of anti-inflammatory factors and preventing the production of pro-inflammatory factors in this study.

The antioxidant capacity reflects the physiological status of aquatic animals [89]. It is well known that the antioxidant enzyme SOD catalyzes the dismutation of the superoxide anion into O_2_ and H_2_O_2_. H_2_O_2_ is subsequently degraded into H_2_O by the antioxidant enzyme GSH-Px in the cytosol. These enzymes are easily induced by oxidative stress and their activity is usually used to reflect the ability of cleaning free radicals in cells [90]. T-AOC reflects the protective capacity of non-enzymatic antioxidant defense system [91]; and MDA as one of the end-products of lipid peroxidation reflects the degree of lipid peroxidation caused by free radicals and indirectly reflects the degree of cell damage [92]. In the present study, the intestinal GSH-Px activity was decreased, and the MDA content was increased in fish fed the high-SBM diets vs. those fed the FM diets. Similarly, fish showed decreased activities of SOD [69,77,93], GSH-Px [48], and T-AOC [77] as well as increased MDA content [69,77] when fed the diets with an FM replacement of high SBM. This indicated that fish suffered from oxidative stress. The decline in the immunity and antioxidant capacity of animals is associated with the presence of ANFs in the SBM [94,95]. After a dietary intervention with LF, fish had higher intestinal GSH-Px activity and lower MDA content vs. with high-SBM diets. Furthermore, the intestinal GSH-Px activity showed an increasing trend with increasing dietary LF inclusion levels and even returned to the level of the diet FM. The findings suggest a dietary LF intervention could promote the intestinal antioxidant capacity and prevent the intestinal lipid peroxidation in juvenile orange-spotted groupers with SBMIE, through influencing both enzymatic and non-enzymatic antioxidants. Consistent with our results, a reduction in the liver MDA content on day 15 and an increase in the liver T-AOC on day 30 were observed after *Aeromonas veronii*-induced Nile tilapias were administrated with LF at 0.8 g/kg diet [96]; increased SOD activity and decreased MDA content in the hepatopancreas were observed in shrimp receiving the diets with 1.5–2.5 g/kg LF [97]; and similar effects occurred for weaned piglets when they were fed diets with LF [37]. The above results validated that a dietary LF intervention has a protective effect against oxidative stress resulting from different sources of stress. However, higher dietary LF inclusion levels did not improve the oxidative stress of the fish species. This was also reflected in the growth, that is, higher dietary LF inclusion levels did not further enhance the growth performance.

Besides the antioxidant capacity, another important indicator to measure the health status of fish is the plasma biochemical components [24,98]. HDL-C participates in the transportation of lipids from peripheral tissues to the liver for catabolism, whereas LDL-C transports cholesterol from the liver to peripheral tissues [22]. High-SBM diets could decrease the plasma TC, TG, LDL-C, and/or HDL-C contents vs. the control (FM diet) in many previous studies [1,69,99,100,101], as observed in our current study. This indicates an inferior nutritional status of fish caused by high-SBM inclusion. Not as expected, plasma contents of TC, HDL-C and LDL-C were still lower in high-SBM diets with LF intervention than that in FM diets. As a result, the dietary intervention of LF on the malnutrition of grouper caused by high-SBM diets is limited, though it improved the overall antioxidant capacity and growth.

## 5. Conclusions

The supplementation of LF in high-SBM diets not only improves the growth performance and intestinal morphology but also reduces the permeability of intestinal mucosal cells and attenuates the intestinal inflammatory response in juvenile orange-spotted groupers, an improvement of the nutritional status under the intervention of LF. The optimal appropriate supplementation level of LF was 5.8 g/kg based on the quadratic regression analysis of the percent weight gain against dietary LF inclusion levels. This is the first report on the intervention effect of dietary LF on grouper enteritis induced by dietary SBM. Our current study will provide a basis for LF use as a functional feed additive to alleviate fish SBMIE.

## Figures and Tables

**Figure 1 metabolites-12-00915-f001:**
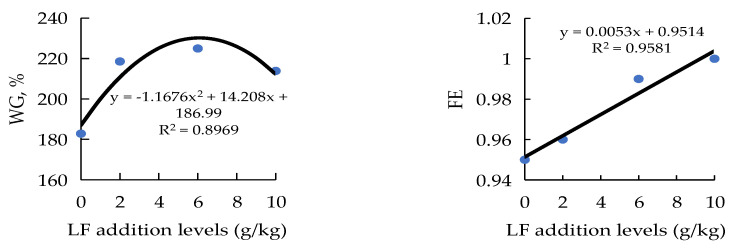
The relationship between weight gain (WG) and feed efficiency (FE) of juvenile orange-spotted groupers and the lactoferrin (LF) inclusion levels in SBM60 diets in a 56-day feeding period. Values are means of 3 triplicates per dietary treatment. SBM60, high-soybean-meal diet with 600 g/kg fish meal protein replacement and without LF supplementation; LF2, LF6, and LF10 were added 2, 6, and 10 g/kg LF in SBM60 diets, respectively.

**Figure 2 metabolites-12-00915-f002:**
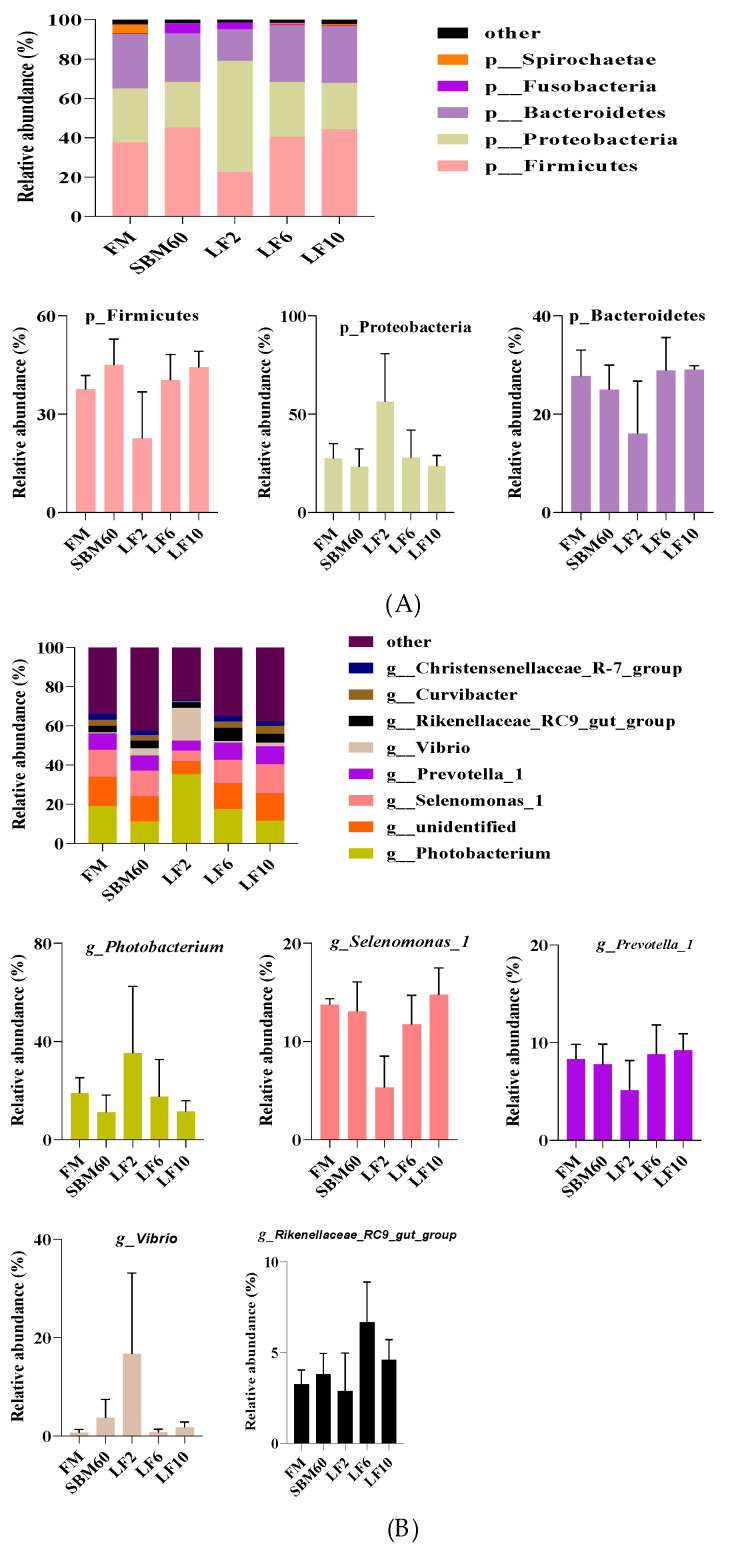
Relative abundances of the dominant bacterial at phylum (**A**) and genus (**B**) in distal intestine (DI) of juvenile orange-spotted grouper fed different diets in a 56-day feeding period. Bars bearing the different letters are significantly different (*p* < 0.05). FM, fish meal diet (control diet); SBM60, high-soybean-meal diet with 600 g/kg fish meal protein replacement and without LF supplementation; LF2, LF6, and LF10 were added 2, 6, and 10 g/kg LF in SBM60 diets, respectively.

**Figure 3 metabolites-12-00915-f003:**
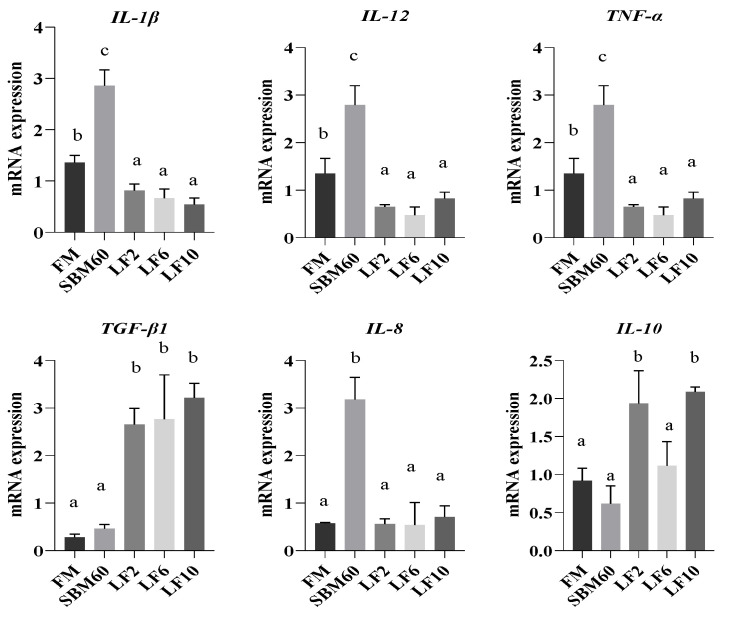
Effect of lactoferrin (LF) inclusion levels in SBM60 diets on mRNA levels of intestinal inflammatory factor genes of juvenile orange-spotted groupers in a 56-day feeding period. Data are presented as means ± SEM (*n* = 3 tanks). Statistical analysis was performed by one-way ANOVA, followed by S–N–K test. Bars bearing the different letters are significantly different (*p* < 0.05). FM, fish meal diet (control diet); SBM60, high-soybean-meal diet with 600 g/kg fish meal protein replacement and without LF supplementation; LF2, LF6, and LF10 were added 2, 6, and 10 g/kg LF in SBM60 diets, respectively. Abbreviations: IL-8, interleukin-8; IL-1β, interleukin-1β; TNF-α, tumor necrosis factor-α; IL-12, interleukin-12; TGF-β1, transforming growth factor-β1; IL-10, interleukin-10.

**Table 1 metabolites-12-00915-t001:** Formulations and nutrient levels of the experimental diets (on an as-fed basis, g/kg).

Items	Diets ^1^				
	FM	SBM60	LF2	LF6	LF10
Ingredients					
Fish meal ^2^	520	220	220	220	220
Casein	119.8	112.7	112.7	112.7	112.7
Gelatin	30	28.2	28.2	28.2	28.2
Soybean meal ^3^	—	470	470	470	470
Soybean oil	35	35	35	35	35
Fish oil	8.2	35.2	35.2	35.2	35.2
Soybean lecithin	20	20	20	20	20
Lactoferrin 4	—	—	2	6	10
Corn starch	177.2	32.6	30.6	26.6	22.6
Sodium alginate	10	10	10	10	10
Ca(H_2_PO_4_)_2_	15	15	15	15	15
Choline chloride	4	4	4	4	4
Stay-C (350 g/kg)	0.3	0.3	0.3	0.3	0.3
Vitamin premix ^5^	4	4	4	4	4
Mineral premix ^5^	5	5	5	5	5
Taurine	5	8	8	8	8
Microcrystalline cellulose	46.5	—	—	—	—
Total	1000	1000	1000	1000	1000
Nutrient level (analyzed values)					
Dry matter	950.6	957.8	955.6	952.7	954.7
Crude protein	480.5	503.4	517.2	513.3	515
Crude lipid	120.6	114.2	116.3	116.4	116.3
Ash	91.7	84.2	84.8	84	82.9

^1^ FM, fish meal diet (control diet); SBM60, high-soybean-meal diet with 600 g/kg fish meal protein replacement and without LF supplementation; LF2, LF6, and LF10 were added 2, 6, and 10 g/kg LF in SBM60 diets, respectively. ^2^ Fish meal was obtained from Austral Group S.A.A.Peru. (crude protein 703.4 g/kg, crude lipid 90.6 g/kg). ^3^ Soybean meal was obtained from Jiakang Feed Co., Ltd., Xiamen, China. (crude protein 466.0 g/kg, crude lipid 7.2 g/kg). ^4^ Lactoferrin (LF, food grade, the purity of lactoferrin was 970 g/kg) was obtained from Fujian Furun Pharmaceutical Co., Ltd. (Fuzhou, China). ^5^ The vitamin and mineral premixes were obtained from Guangzhou Feixite Aquatic Technology Co., Ltd (Guangzhou, China). Vitamin premix (per kg diet): VA, 10 mg; VD, 10 mg; VE, 100 mg; VB1, 10 mg; VB2, 20 mg; VB6, 20 mg; VB12, 0.05 mg; nicotinic acid, 50 mg; calcium-D-pantothenate, 100 mg; D-biotin, 1 mg; meso-inositol, 500 mg; folic acid, 4 mg. Mineral premix (per kg diet): ferric citrate, 497 mg; CuSO_4_·5H_2_O, 24 mg; ZnSO_4_·7H_2_O, 176 mg; MnSO_4_·4H_2_O, 122 mg; CoCl_2_·6H_2_O, 0.18 mg; KIO_3_, 0.51 mg; Na_2_SeO_3_, 0.33 mg.

**Table 2 metabolites-12-00915-t002:** Primers sequences for lipid-related genes and reference genes used for real-time PCR of juvenile orange-spotted groupers.

Genes ^1^	Primer Sequence (5′ to 3′) ^2^	E-Value (%)	Accession Number
*IL-8*	F: AAGTTTGCCTTGACCCCGAA	94.0	FJ913064.1
	R: TGAAGCAGATCTCTCCCGGT		
*IL-1β*	F: GCAACTCCACCGACTGATGA	116.0	EF582837.1
	R: ACCAGGCTGTTATTGACCCG		
*IL-10*	F: GTCCACCAGCATGACTCCTC	99.0	KJ741852.1
	R: AGGGAAACCCTCCACGAATC		
*TGF-β1*	F: GCTTACGTGGGTGCAAACAG	102.0	GQ503351.1
	R: ACCATCTCTAGGTCCAGCGT		
*IL-12*	F: CCAGATTGCACAGCTCAGGA	115.0	KC662465.1
	R: CCGGACACAGATGGCCTTAG		
*TNF-α*	F: GGATCTGGCGCTACTCAGAC	91.0	FJ009449.1
	R: CGCCCAGATAAATGGCGTTG		
*β-actin*	F: TGCTGTCCCTGTATGCCTCT	104.0	AY510710.2
	R: CCTTGATGTCACGCACGAT		

^1^ IL-8, interleukin-8; IL-1β, interleukin-1β; TNF-α, tumor necrosis factor-α; IL-12, interleukin-12; TGF-β1, transforming growth factor-β1; IL-10, interleukin-10. ^2^ F, forward; R, reverse.

**Table 3 metabolites-12-00915-t003:** Effects of lactoferrin (LF) supplementation in high-SBM diets on growth performance and whole-body proximate composition of juvenile orange-spotted groupers in a 56-day feeding period ^1^.

Items	Diets ^2^				
	FM	SBM60	LF2	LF6	LF10
Growth performance					
IBW (g/fish) ^3^	33.82 ± 0.10	33.76 ± 0.06	33.74 ± 0.05	33.80 ± 0.07	33.98 ± 0.01
FBW (g/fish) ^3^	113.24 ± 0.66 ^b^	95.47 ± 2.59 ^a^	107.49 ± 4.28 ^b^	109.85 ± 3.46 ^b^	106.65 ± 2.96 ^b^
WG (%) ^3^	234.81 ± 2.54 ^b^	182.81 ± 7.24 ^a^	218.56 ± 12.96 ^b^	224.98 ± 9.89 ^b^	208.68 ± 5.73 ^ab^
SGR (%/d) ^3^	2.16 ± 0.01 ^b^	1.86 ± 0.05 ^a^	2.07 ± 0.07 ^b^	2.10 ± 0.05 ^b^	2.04 ± 0.05 ^b^
FE ^3^	0.98 ± 0.00	0.95 ± 0.00	0.96 ± 0.12	0.99 ± 0.07	1.00 ± 0.12
Survival (%) ^3^	100.00 ± 0.00	97.78 ± 1.11	100.00 ± 0.00	100.00 ± 0.00	98.89 ± 1.11
HSI (%) ^4^	1.31 ± 0.09	1.24 ± 0.04	1.28 ± 0.07	1.24 ± 0.05	1.17 ± 0.02
CF (g/cm^3^) ^4^	3.16 ± 0.07	3.05 ± 0.11	3.19 ± 0.02	2.93 ± 0.12	2.94 ± 0.03
Proximate composition (%)					
Moisture	67.05 ± 0.21	67.27 ± 0.22	67.56 ± 0.37	67.42 ± 0.34	68.26 ± 0.39
Crude protein	18.01 ± 0.49	17.95 ± 0.27	17.93 ± 0.90	19.20 ± 0.33	17.75 ± 0.42
Crude lipid	8.25 ± 0.17	7.90 ± 0.29	8.18 ± 0.40	7.85 ± 0.32	7.49 ± 0.11
Ash	5.00 ± 0.15	4.96 ± 0.07	4.90 ± 0.06	4.87 ± 0.22	4.93 ± 0.09

Abbreviations: IBW, initial body weight (g/fish); FBW, final body weight (g/fish); FI, feed intake (g/fish); FN, final number; IN, initial number; LW, liver weight (g/fish); BW, body weight (g/fish); BL, body length (cm/fish). ^1^ Statistical analysis was performed by one-way ANOVA, followed by S–N–K test. ^2^ FM, fish meal diet (control diet); SBM60, high-soybean-meal diet with 600 g/kg fish meal protein replacement and without LF supplementation; LF2, LF6, and LF10 were added 2, 6, and 10 g/kg LF in SBM60 diets, respectively. ^3^ Values are presented as the means ± SEM (*n* = 3 tanks). ^4^ Values are presented as the means ± SEM (*n* = 27 fish). ^a,b^ Values in the same row with different superscripts indicate significant differences (*p* < 0.05), while that with the same letter or no letter superscripts indicate no significant differences (*p* > 0.05). WG, weight gain (%) = 100 × (FBW − IBW)/IBW. SGR, specific growth rate (%/d) = 100 × (lnFBW − lnIBW)/days. FE, feed efficiency = 100 × (FBW − IBW)/FI (as fed basis, g/fish). Survival (%) = 100 × FN/IN. HSI, hepatosomatic index (%) = 100 × LW/BW. CF, condition factor (g/cm^3^) = 100 × BW/(BL)^3^.

**Table 4 metabolites-12-00915-t004:** Effect of lactoferrin (LF) supplementation in high-SBM diets on intestinal antioxidant indices of juvenile orange-spotted groupers in a 56-day feeding period ^1^.

Items ^3^	Diets ^2^				
	FM	SBM60	LF2	LF6	LF10
SOD (U/mg protein)	71.67 ± 5.50	68.92 ± 1.76	62.16 ± 5.85	64.26 ± 3.23	60.70 ± 2.15
GSH-Px (U/mg protein)	79.58 ± 3.31 ^bc^	65.29 ± 2.97 ^a^	82.52 ± 1.76 ^bc^	72.99 ± 1.35 ^b^	86.76 ± 4.00 ^c^
T-AOC (U/mg protein)	0.19 ± 0.01	0.19 ± 0.01	0.21 ± 0.01	0.19 ± 0.02	0.18 ± 0.01
MDA (nmol/mg protein)	3.00 ± 0.28 ^a^	4.56 ± 0.88 ^b^	1.97 ± 0.21 ^a^	1.86 ± 0.12 ^a^	2.55 ± 0.05 ^a^
Y_MDA_ = 0.0783X^2^ − 0.9351X + 4.2006, R^2^ = 0.8297, X = LF supplementation levels (g/kg)					

^1^ Data were presented as means ± SEM (*n* = 3 tanks). Statistical analysis was performed by one-way ANOVA, followed by S–N–K test. ^2^ FM, fish meal diet (control diet); SBM60, high-soybean-meal diet with 600 g/kg fish meal protein replacement and without LF supplementation; LF2, LF6, and LF10 were added 2, 6, and 10 g/kg LF in SBM60 diets, respectively. ^3^ Abbreviations: T-AOC, total antioxidant capacity; SOD, superoxide dismutase; MDA, malondialdehyde; GSH-Px, glutathione peroxidase. ^a^^–^^c^ Values in the same row with different superscripts indicate significant differences (*p* < 0.05), while that with the same letter or no letter superscripts indicate no significant differences (*p* > 0.05).

**Table 5 metabolites-12-00915-t005:** Effect of lactoferrin (LF) supplementation in high-SBM diets on plasma components of juvenile orange-spotted groupers in a 56-day feeding period ^1^.

Items ^3^	Diets ^2^				
	FM	SBM60	LF2	LF6	LF10
HDL-C (mmol/L)	1.06 ± 0.05 ^b^	1.00 ± 0.03 ^b^	0.81 ± 0.05 ^a^	1.03 ± 0.03 ^b^	0.83 ± 0.09 ^a^
LDL-C (mmol/L)	0.28 ± 0.01 ^c^	0.19 ± 0.01 ^b^	0.18 ± 0.01 ^b^	0.12 ± 0.01 ^a^	0.12 ± 0.01 ^a^
TC (mmol/L)	3.77 ± 0.21 ^b^	3.49 ± 0.23 ^ab^	3.35 ± 0.19 ^ab^	3.09 ± 0.09 ^ab^	2.90 ± 0.09 ^a^
TG (mmol/L)	1.61 ± 0.17	1.36 ± 0.08	1.25 ± 0.06	1.57 ± 0.10	1.55 ± 0.07
Y_LDL-C_ = −0.0079X + 0.188, R^2^ = 0.8573, X = LF supplementation levels (g/kg) Y_TC_ = −0.0592x + 3.4741, R^2^ = 0.9931, X = LF supplementation levels (g/kg)					

^1^ Data were presented as means ± SEM (*n* = 3 tanks). Statistical analysis was performed by one-way ANOVA, followed by S–N–K test. ^2^ FM, fish meal diet (control diet); SBM60, high-soybean-meal diet with 600 g/kg fish meal protein replacement and without LF supplementation; LF2, LF6, and LF10 were added 2, 6, and 10 g/kg LF in SBM60 diets, respectively. ^3^ Abbreviations: TG, triglyceride; TC, total cholesterol; HDL-C, high-density lipoprotein cholesterol; LDL-C, low-density lipoprotein cholesterol. ^a^^–^^c^ Values in the same row with different superscripts indicate significant differences (*p* < 0.05), while that with the same letter or no letter superscripts indicate no significant differences (*p* > 0.05).

**Table 6 metabolites-12-00915-t006:** Effect of lactoferrin (LF) supplementation in high-SBM diets on activities of intestinal digestive enzymes of juvenile orange-spotted groupers in a 56-day feeding period ^1^.

Items	Diets ^2^				
	FM	SBM60	LF2	LF6	LF10
Lipase (U/mg protein)	0.68 ± 0.00 ^a^	0.61 ± 0.00 ^a^	0.84 ± 0.05 ^b^	0.86 ± 0.02 ^b^	0.95 ± 0.05 ^bc^
Amylase (U/mg protein)	0.76 ± 0.06	0.73 ± 0.11	0.90 ± 0.04	0.73 ± 0.07	0.82 ± 0.06
Trypsin (U/g protein)	256.07 ± 17.23 ^b^	175.55 ± 17.55 ^a^	238.95 ± 17.46 ^b^	235.03 ± 9.36 ^b^	283.57 ± 8.83 ^bc^
Protease (U/mg protein)	20.54 ± 0.87	15.91 ± 2.04	17.37 ± 2.91	23.54 ± 2.56	26.54 ± 2.53
Y_Lipase_ = 0.0283X + 0.6876, R^2^ = 0.7515, X = LF supplementation levels (g/kg) Y_Trypsin_ = 8.8954X + 193.25, R^2^ = 0.7917, X = LF supplementation levels (g/kg) Y_Protease_ = 1.1231X + 15.786, R^2^ = 0.9775, X = LF supplementation levels (g/kg)					

^1^ Data were presented as means ± SEM (*n* = 3 tanks). Statistical analysis was performed by one-way ANOVA, followed by S–N–K test. ^2^ FM, fish meal diet (control diet); SBM60, high-soybean-meal diet with 600 g/kg fish meal protein replacement and without LF supplementation; LF2, LF6, and LF10 were added 2, 6, and 10 g/kg LF in SBM60 diets, respectively. ^a^^–^^c^ Values in the same row with different superscripts indicate significant differences (*p* < 0.05), while that with the same letter or no letter superscripts indicate no significant differences (*p* > 0.05).

**Table 7 metabolites-12-00915-t007:** Effect of lactoferrin (LF) supplementation in high-SBM diets on the biochemical indices of intestinal mucosal permeability of orange-spotted groupers in a 56-day feeding period ^1^.

Items ^3^	Diets ^2^				
	FM	SBM60	LF2	LF6	LF10
DAO (U/L)	19.75 ± 1.39	20.59 ± 1.05	16.09 ± 1.22	17.77 ± 2.40	17.72 ± 1.47
D-Lac (nmol/mL)	2.03 ± 0.20 ^a^	4.05 ± 0.23 ^b^	1.41 ± 0.18 ^a^	1.59 ± 0.21 ^a^	1.41 ± 0.07 ^a^
ET-1 (ng/L)	1.91 ± 0.07	2.12 ± 0.09	2.27 ± 0.10	1.92 ± 0.13	2.36 ± 0.20
ET (EU/L)	1.51 ± 0.03 ^b^	1.70 ± 0.10 ^c^	1.23 ± 0.01 ^a^	1.25 ± 0.01 ^a^	1.25 ± 0.04 ^a^

^1^ Data were presented as means ± SEM (*n* = 3 tanks). Statistical analysis was performed by one-way ANOVA, followed by S–N–K test. ^2^ FM, fish meal diet (control diet); SBM60, high-soybean-meal diet with 600 g/kg fish meal protein replacement and without LF supplementation; LF2, LF6, and LF10 were added 2, 6, and 10 g/kg LF in SBM60 diets, respectively. ^3^ Abbreviations: DAO, diamine oxidase; D-Lac, d-lactic acid; ET, endotoxin; ET-1, endothelin-1. ^a^^–^^c^ Values in the same row with different superscripts indicate significant differences (*p* < 0.05), while that with the same letter or no letter superscripts indicate no significant differences (*p* > 0.05).

**Table 8 metabolites-12-00915-t008:** Effect of lactoferrin (LF) supplementation in high-SBM diets on the intestinal morphology of juvenile orange-spotted groupers in a 56-day feeding period ^1^.

Items ^3^	Diets ^2^				
	FM	SBM60	LF2	LF6	LF10
PI					
lMF (μm)	577.30 ± 87.68	489.10 ± 54.31	574.92 ± 35.62	513.26 ± 50.67	737.53 ± 95.20
tML (μm)	63.24 ± 6.74	64.56 ± 8.11	79.05 ± 2.27	86.60 ± 9.31	86.00 ± 2.51
nMF (unit)	42.50 ± 4.25	45.83 ± 3.09	51.67 ± 1.36	50.33 ± 8.62	48.00 ± 3.55
MI					
lMF (μm)	465.12 ± 50.20 ^ab^	381.90 ± 42.42 ^a^	580.47 ± 9.06 ^b^	356.66 ± 9.37 ^a^	540.48 ± 47.06 ^b^
tML (μm)	53.53 ± 2.44 ^ab^	44.96 ± 4.06 ^a^	76.61 ± 7.02 ^b^	63.13 ± 4.61 ^ab^	69.62 ± 7.81 ^b^
nMF (unit)	34.33 ± 2.20	31.67 ± 1.01	43.00 ± 4.36	34.83 ± 3.49	39.17 ± 0.33
DI					
lMF (μm)	417.87 ± 63.72	337.13 ± 44.48	437.82 ± 22.32	397.03 ± 4.38	466.67 ± 53.64
tML (μm)	87.58 ± 7.61 ^b^	51.53 ± 1.48 ^a^	74.80 ± 3.34 ^b^	86.49 ± 1.35 ^b^	69.78 ± 7.20 ^b^
nMF (unit)	32.00 ± 5.20	37.00 ± 4.00	40.83 ± 5.33	35.00 ± 1.04	34.00 ± 3.21
DI: Y_tML_ = −1.0492X^2^ + 12.167X + 52.619, R^2^ = 0.9884, X = LF supplementation levels (g/kg)					

^1^ Data were presented as means ± SEM (*n* = 3 tanks). Statistical analysis was performed by one-way ANOVA, followed by S–N–K test. ^2^ FM, fish meal diet (control diet); SBM60, high-soybean-meal diet with 600 g/kg fish meal protein replacement and without LF supplementation; LF2, LF6, and LF10 were added 2, 6, and 10 g/kg LF in SBM60 diets, respectively. ^3^ Abbreviations: PI, proximal intestine; MI, middle intestine; DI, distal intestine; nMF, mucosal fold number; tML, muscle layer thickness; lMF, mucosal fold length. ^a,b^ Values in the same row with different superscripts indicate significant differences (*p* < 0.05), while that with the same or no letter superscripts indicate no significant differences (*p* > 0.05).

## Data Availability

The data that support the findings of this study are available from the corresponding author upon reasonable request.

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
