# Peer review of "Dietary Lactoferrin Supplementation Improves Growth Performance and Intestinal Health of Juvenile Orange-Spotted Groupers (Epinephelus coioides)"

_metabolites, 2022, doi:10.3390/metabo12100915_

Round 1

Reviewer 1 Report

The paper written by Song et al. presents effects of dietary lactoferrin (LF) on growth and physiological condition of orange spotted grouper, that would provide novel information of LF's effects on fish and aquaculture. However, the paper need several revisions as commented below.

1.      Update references overall the manuscript. A number of paper have been published on the relation between dietary LF and fish performances including a review (Luna-Castro et al., 2021, https://doi.org/10.1111/are.15621).

2.       A study, which focused on the effects of LF for orange spotted grouper (Yokoyama et al. 2006, https://doi.org/10.1016/j.aquaculture.2005.12.001), suggested dietary LF has no effect on growth enhancement. This is also opposite results from that in the present study. Please add this reference and discuss this issue in the discussion part.

3.      Provide the purity of LF used in the present study (Table 1).

4.      Present test diets were pelletized using an extruder (Page 3, line 11-13), probably be exposed to high temperature and high pressure. The reviewer concerns high temperature and high pressure condition result in denaturing and loosing the activity of LF. Please discuss this issue.

5.      Please describe analytical method for digestive enzymes activity in the materials and methods section.

6.      What is difference between 'trypsin' and 'protease' (Table 6)? Is trypsin a part of protease isn't it, ? If so, why the trypsin activity is higher than the protease activity?

Reviewer 2 Report

The manuscript describes the effect of dietary lactoferrin administration on the growth performance and intestinal health of Epinephelus coioides fed high-soybean meal diets. The topic is definitely worth investigating and has clear benefits to aquafeeds. The manuscript is generally well-written. The experimental design was sound, and the workload was adequate. The discussion is appropriate and the authors made an admirable attempt in explaining the intervention mechanism of lactoferrin on soybean meal induced enteritis of fish. The manuscript is recommended to be published in this journal after minor revisions.

Minor stylistic errors are displayed as follows

The abstract needs to be more concise.

Line 121 and line 126: A wrong dosage unit for the anesthetic eugenol. Please confirm which anesthetic was used in this study.

Line 601: Correct the number in parentheses (× 10-4) as the correct format (× 10-4) .

Line 672, line 698, line 759, line 641: There is inconsistent expression of the same magazine name.

Line 715, line 720: The format of this reference magazine name does not meet the requirement. Please check other places.

Line 776: page number missing.

Line 703 and line 775: The bacteral names should be change to italics. Please check other places.

Line 729 and line 793: “Silvery-Black Porgy” and “Gilthead sea bream” are not a proper noun, they should be change to “silvery-black porgy” and “gilthead sea bream”. Please check others.

Round 2

Reviewer 1 Report

The present paper has been much improved from the previous version. It looks that the manuscript is now ready for publishing.
